# Hormone Replacement Therapy and Risks of Various Cancers in Postmenopausal Women with De Novo or a History of Endometriosis

**DOI:** 10.3390/cancers16040809

**Published:** 2024-02-16

**Authors:** Hee Joong Lee, Banghyun Lee, Hangseok Choi, Minkyung Lee, Kyungjin Lee, Tae Kyoung Lee, Sung Ook Hwang, Yong Beom Kim

**Affiliations:** 1Department of Obstetrics & Gynecology, Uijeongbu St. Mary’s Hospital, College of Medicine, The Catholic University of Korea, Seoul 11765, Republic of Korea; 2Department of Obstetrics and Gynecology, Inha University Hospital, Inha University College of Medicine, Incheon 22332, Republic of Korea; s0164802536@naver.com (M.L.); tklee@inha.ac.kr (T.K.L.);; 3Medical Science Research Center, Korea University College of Medicine, Seoul 02841, Republic of Korea; neuldol@gmail.com; 4Department of Obstetrics and Gynecology, Seoul National University Bundang Hospital, Seongnam-si 13620, Republic of Korea; ybkimlh@snubh.org

**Keywords:** breast cancer, endometriosis, hormone replacement therapy, liver cancer, lung cancer, thyroid cancer, uterine cancer, postmenopause

## Abstract

**Simple Summary:**

Hormone replacement therapy (HRT) is used to improve the climacteric symptoms in postmenopausal women with a history of endometriosis. However, the impact of HRT on the malignant transformations of postmenopausal endometriosis is unclear. Therefore, this study examined the effects of HRT on the occurrences of various cancers in postmenopausal women with de novo or a history of endometriosis using the data from the Korean nationwide cohort study. In this study, the use of HRT increased the risk of uterine cancer, but it did not increase the risks of another nine cancers. Interestingly, the use of combined estrogen and progesterone reduced the risks of liver and thyroid cancers, and the use of estrogen alone reduced the risks of breast and lung cancers. Tibolone use was not associated with the risks of ten cancers. These results provide useful information on the use of HRT in women with de novo or a history of endometriosis.

**Abstract:**

This study examined the impact of hormone replacement therapy (HRT) on the occurrence of various cancers in postmenopausal women with de novo or a history of endometriosis. In the datasets for ten cancers (cervical, uterine, ovarian, breast, colon, gastric, liver, lung, pancreatic, and thyroid), women who received HRT (the HRT group) and those who did not (the control group) were selected by a 1:1 matching with those who met the study criteria. In the dataset for each cancer, the incidence of each cancer was very low (0.2% to 1.5% in the HRT group and 0.2% to 1.3% in the control group). The duration of HRT was 1.3 ± 2.1 years. After adjusting for co-variables, HRT was a significant risk factor for uterine cancer (*p* < 0.05). However, the risk of liver cancer decreased significantly with duration of HRT (*p* < 0.05). Moreover, combined estrogen and progesterone decreased the risks of liver and thyroid cancers significantly (*p* < 0.05), and estrogen alone decreased the risks of breast and lung cancers significantly (*p* < 0.05). Tibolone was not associated with the risk of any of the cancers assessed. These results can help guide the use of HRT in women with de novo or a history of endometriosis.

## 1. Introduction

Endometriosis is a chronic, recurrent, benign disease involving the presence of endometrial tissue outside the uterine cavity. Endometriosis occurs mostly in the pelvic cavity, but extrapelvic endometriosis can occur, albeit rarely, in distant sites from gynecological organs, depending on the location of endometrial tissue implantation [1].

Endometriosis affects women from premenarche to postmenopause [2,3]. The incidence of endometriosis is 6–10% and 2–4% in premenopausal and postmenopausal women, respectively [4]. The risk of a malignant transformation of endometriotic lesions is approximately 1% [4,5,6]. Endometriosis recurrence or malignant transformations can occur in the postmenopausal period [2,7,8].

Many case reports and series have revealed the recurrence or malignant transformation of endometriosis after hormone replacement therapy (HRT), particularly with estrogen alone in postmenopausal women with a history of endometriosis [2,9,10,11,12,13]. In contrast, one randomized controlled trial (RCT) (*n* = 172) reported that HRT did not increase the recurrence of endometriosis significantly in postmenopausal women with a history of endometriosis [14]. Moreover, in one retrospective study (*n* = 93), estrogen alone showed a trend toward an increased risk of a malignant transformation of endometrioma [15]. In addition, in one RCT (*n* = 21), tibolone showed a better pelvic pain rate compared to estrogen with or without progesterone in women with residual pelvic endometriosis, but the difference was not significant [16]. Therefore, the impact of HRT on the risks of endometriosis recurrence and malignant transformation in postmenopausal women with a history of endometriosis is unclear owing to the lack of high-quality studies [2]. On the other hand, based on low-quality evidence, combined estrogen and progesterone are currently recommended to improve menopausal symptoms in postmenopausal women with a history of endometriosis [9,17]. Tibolone can also be considered [9].

The relationships between HRT and the malignant transformation of de novo or recurrent postmenopausal endometriosis are still unclear. Moreover, the malignant transformations of endometriosis might occur in pelvic and extrapelvic endometriosis. Therefore, this nationwide cohort study examined the effects of HRT on the occurrences of various cancers in postmenopausal women with de novo or a history of endometriosis using Korean health insurance review and assessment service (HIRA) data.

## 2. Materials and Methods

The National Health Service, a universal health coverage system in the Republic of Korea, covers ~98% of the population [18]. Most of the National Health Insurance Service data are shared by the HIRA. Patients with cancer receive additional medical payment discounts. Hence, the diagnosis codes for cancer in medical information are highly accurate. This study evaluated the claims data of women diagnosed with endometriosis according to the diagnostic codes in the HIRA between 1 January 2007 and 31 December 2022. The Institutional Review Board of The Catholic Medical Center at the Catholic University of Korea (No. UC23ZISI0019) approved this study on 9 March 2023. The HIRA dataset uses anonymous identification codes to protect personal information, as the Bioethics and Safety Act of South Korea requires. Therefore, the requirement for informed consent for the patients included in the data was waived.

This study used the International Classification of Diseases 10th revision (ICD-10), Korea Health Insurance Medical Care Expenses (2017 and 2019 versions), and HIRA Drug Ingredients Codes for the selection and analysis of subjects. Women with endometriosis were defined as having the diagnostic codes for endometriosis (N80x) with the surgery codes within 60 days before or after an initial diagnostic code. Women who received HRT (HRT group) were defined as women who received HRT prescriptions for ≥28 days, whereas those who did not (control group) were defined as women who did not receive a prescription for HRT. The following exclusion criteria were used to select eligible patients: women ≤ 49 years old at the last clinic visit; women diagnosed with endometriosis during the washout period (between 1 January 2007 and 31 December 2007); women diagnosed with each cancer before the date of the first diagnostic code for endometriosis. Women with or without HRT were identified using 1:1 matching according to the age at the last clinic visit in the dataset for each cancer. Women diagnosed with each cancer before receiving HRT and those diagnosed with each cancer within one year of HRT receipt were excluded from the HRT group in the dataset for each cancer. An additional exclusion criterion was women diagnosed with each cancer before menopause or within one year after menopause. The date HRT was started in the HRT group was considered the date of menopause. The date of menopause in the control group was defined by matching with the HRT group. Accordingly, this study excluded women diagnosed with each cancer in the HRT or control groups before HRT or menopause or within one year after HRT or menopause (Figure 1 and Appendix A file). The dataset for each cancer was independently created and not associated with datasets for other cancers. Therefore, women diagnosed with each cancer during the creation of the dataset for each cancer and those diagnosed with each cancer from the dataset for each cancer were candidates for datasets for other cancers.

Women with each cancer were defined as having one or more diagnostic codes for the following cancers: cervical cancer (C53x), uterine cancer (C54x), ovarian cancer (C56x), breast cancer (C50x), colon cancer (C18x), gastric cancer (C16x), liver cancer (C22x), lung cancer (C33 and C34x), pancreatic cancer (C25), and thyroid cancer (C73). Bladder cancer (C67.x), bone cancer (C41.x), brain cancer (C71.x), esophageal cancer (C15.x), leukemia (C91.x, C92.x, C93.x, C94.x, and C95.x), melanoma (C43.x), mesothelioma (C45.x), non-Hodgkin lymphoma (C82.x, C83.x, C84.x, C85.x, and C86.x), oral cancer (C05.x and C06.x), and renal cancer (C64.0 and C65.0) were excluded from the analysis because the incidence of cancer in eligible dataset for each cancer was zero or extremely low (0.05% to 0.09%). The medical insurance type was defined as low socioeconomic status (SES) when it was medical protection. The Charlson Comorbidity Index (CCI) was obtained using the diagnostic code from the date of the last clinic visit to the year before [19]. The methods of surgery for endometriosis included ovarian cystectomy, bilateral or unilateral salpingo-oophorectomy (BSO or USO), hysterectomy, or fulguration. Surgery was defined as the codes for those surgeries and a concurrent diagnostic code for endometriosis or surgery codes within 60 days before or after an initial diagnostic code for endometriosis. Hormone therapy was defined using prescription codes, such as combined estrogen and progesterone, estrogen alone, or tibolone.

### Statistical Analyses

In the HRT and control groups of the dataset for each cancer, women were considered homogenous if they were the same age at the last clinic visits. Therefore, the two groups were 1:1 matched for age. Logistic regression analysis, adjusted for the confounding factors or not, was conducted to determine the relationships between the independent risk factors and each group in the dataset for each cancer. In the dataset for each cancer, the association between the two groups and time-to-event data was analyzed using the Kaplan–Meier and Cox proportional-hazards regression models. Cox multiple regression was used to control for confounding variables. An independent t-test was used to analyze the continuous variable in the dataset for each cancer. All statistical analyses were conducted using two-tailed tests, and a *p* value < 0.05 was considered significant. SAS version 9.4 (SAS Institute Inc., Cary, NC, USA) was used to explore and modify big data, and the analysis was conducted using R version 3.5.1 (R Foundation for Statistical Computing, Vienna, Austria) [20,21].

## 3. Results

The data of 135,379 women with a diagnostic code for endometriosis first registered by the HIRA between 2007 and 2022 were extracted in the initial study. Women in this group who met the study eligibility criteria were independently selected in ten datasets for ten cancers (Figure 1). Furthermore, in the dataset for each cancer, these were assigned by 50.0% to the HRT and control groups, respectively.

### 3.1. Characteristics of Datasets for Various Cancers

Table 1 and Appendix A list the baseline characteristics of the datasets for the ten cancers.

The mean age at the last clinic visit in both groups was 55.5 ± 4.9 years in the dataset for each cancer (Appendix A). In the dataset for each cancer, the mean ages at the endometriosis diagnosis ranged from 48.4 ± 6.1 to 48.5 ± 6.2 years and were significantly younger in the HRT group than in the control group (*p* < 0.001) (Appendix A). The rates of hysterectomy for benign disease were significantly lower in the HRT group than in the control group in the dataset for each cancer (*p* < 0.001) (Appendix A). In the dataset for each cancer, among the methods of surgery for endometriosis, the rates of ovarian cystectomy, BSO, or USO were significantly higher in the HRT group than in the control group. By contrast, the rate of hysterectomy was significantly lower in the HRT group than in the control group (*p* < 0.05) (Appendix A). The number of surgeries for endometriosis was significantly higher in the HRT group than in the control group in the dataset for each cancer (*p* < 0.05) (Appendix A). The mean duration of HRT was 1.3 ± 2.1 years in the dataset for each cancer (Table 1). Combined estrogen and progesterone, estrogen alone, and tibolone were used in approximately 34.7%, 48.8%, and 51.4% of the HRT group, respectively, in the dataset for each cancer (Table 1). In the dataset for each cancer, the rate of HRT use before and after the endometriosis diagnosis was approximately 26.0% and 74.0%, respectively (Appendix A). Appendix A lists the SES and CCI at the last clinic visit and the distribution of patients according to the year of endometriosis diagnosis in the dataset for each cancer.

### 3.2. Incidences of Various Cancers

The incidence of each cancer was very low (0.2% to 1.5% in the HRT group and 0.2% to 1.3% in the control group). The receipt of HRT was not significantly associated with the incidence of each cancer (Table 2).

### 3.3. Risk Factors for Various Cancers

Multivariate analysis adjusted for the potential confounding variables revealed the following results. The risks of cervical, ovarian, colon, gastric, and lung cancers increased significantly with age at endometriosis diagnosis in the total population, the HRT group, and/or the control group (*p* < 0.05). By contrast, the risk of breast cancer decreased significantly with age at endometriosis diagnosis in the total population and the control group (*p* < 0.05). In addition, other cancers did not show associations with age at endometriosis diagnosis. Regarding the year of endometriosis diagnosis, the risks of cervical, uterine, and ovarian cancers increased significantly with time in the total population and both groups (*p* <0.001), whereas the risks of colon, gastric, liver, and lung cancers decreased significantly with time in total population, the HRT group, and/or the control group (*p* < 0.05). In addition, other cancers did not show associations with the year of an endometriosis diagnosis. The risks of uterine, ovarian, and thyroid cancers significantly increased as the number of surgeries for endometriosis increased in the total population, the HRT group, and/or the control group (*p* < 0.05), but the other cancers did not show associations with the number of surgeries for endometriosis. The risks of cervical, uterine, and ovarian cancers significantly decreased in women who had undergone a hysterectomy for benign disease in the total population, and both groups (*p* < 0.05), but the other cancers assessed did not show associations with a hysterectomy for benign disease (Appendix A).

The receipt of HRT increased the risk of uterine cancer significantly (*p* < 0.05), but it was not associated with the risks of other cancers. According to the increase in HRT duration, the risk of liver cancer decreased significantly (*p* < 0.05), but other cancers did not show associations with the HRT duration. The receipt of combined estrogen and progesterone was significantly associated with a decrease in the risks of liver and thyroid cancers (*p* < 0.05), and the use of estrogen alone was significantly associated with a decrease in the risks of breast and lung cancers (*p* < 0.05). On the other hand, they were not associated with the risks of other cancers. The receipt of tibolone was not associated with the risk for each cancer (Table 3).

### 3.4. Characteristics of Various Cancers

In cervical and thyroid cancers, the times between the endometriosis diagnosis and cancer diagnosis were significantly shorter in the HRT group than in the control group. In contrast, in other cancers, those were similar in the HRT and control groups (Appendix A). The mean times from HRT commencement to each cancer diagnosis ranged from 5.1 to 6.9 years (Appendix A). The mean duration of HRT in women with each cancer ranged from 0.9 to 2.7 years (Appendix A). The mean ages at each cancer diagnosis were significantly younger in the HRT group than in the control group (Appendix A).

## 4. Discussion

In this study, postmenopausal women with a de novo or a history of endometriosis administered or not administered HRT had a low and similar incidence of the ten cancers evaluated. HRT increased the risk of uterine cancer according to multivariate analysis adjusted for confounding factors. Moreover, an increase in the duration of HRT reduces the risk of liver cancer. Moreover, the use of combined estrogen and progesterone reduced the risks of liver and thyroid cancers, and the use of estrogen alone reduced the risks of breast and lung cancers. Furthermore, the use of tibolone was not associated with the risks for ten cancers.

The average age of menopause generally ranges from ~50 and 51 years [22,23]. Therefore, this study excluded women aged ≤49 years at the last clinic visit. HRT is generally used to improve the quality of life in symptomatic postmenopausal women. In the present dataset for each cancer, based on the time sequence between endometriosis diagnosis and beginning of HRT, de novo postmenopausal endometriosis was observed in 26.0% of the women with HRT and a history of endometriosis was noted in 74.0% (Appendix A).

In large-scale cohorts and a pooled analysis, endometriosis and HRT were associated with an increased risk of endometrial, ovarian, and breast cancers [24,25,26]. Moreover, HRT was associated with decreased risks of liver and colon cancers and did not influence gastric and pancreatic cancers [26]. Therefore, this study expected that postmenopausal women with de novo or a history of endometriosis who received HRT would show similar results to previous studies in relation to the risks of the above cancers compared to those who did not receive HRT. On the other hand, HRT increased the risk of only uterine cancer (Table 3).

In a large-scale cohort and meta and pooled analyses, the risks of endometrial cancer (in users of sequential-combined HRT with synthetic progestins) and ovarian cancer increased significantly with the duration of HRT (especially for durations of ≥10 years), but the use of estrogen alone for one to <10 years in women >50 years was not associated with the risk of ovarian cancer [27,28,29]. In the present study, the HRT duration was not a risk factor for endometrial and ovarian cancers in postmenopausal women with a de novo or a history of endometriosis, suggesting that a very short duration of HRT can be attributed. Instead, the risk of liver cancer decreased with the duration of HRT (Table 3). In the dataset for each cancer, the overall mean duration of HRT was similar to that in women with cancer (Table 1 and Appendix A). Each cancer developed a mean of 3.8 to 5.6 years after stopping HRT (Table 1 and Appendix A).

Based on previous studies and the ESHRE guidelines, combined estrogen and progesterone are recommended for symptomatic postmenopausal women with a history of endometriosis. On the other hand, estrogen alone is not recommended because of the high risk of malignant transformation [9,17]. In addition, tibolone is considered a safe hormonal treatment for postmenopausal women with a history of endometriosis [9,16]. In a nationwide cohort study (*n* = 290,186) analyzing HRT users, combined estrogen and progesterone were associated with increased risks of ovarian and breast cancers and a decreased risk of colon cancer. They did not influence endometrial, liver, gastric, and pancreatic cancers [26]. Moreover, estrogen alone and tibolone were associated with an increased risk of endometrial and breast cancers and a decreased risk of colon cancer. In addition, they did not influence ovarian (estrogen alone and tibolone), liver (estrogen alone), gastric (estrogen alone), and pancreatic cancers (estrogen alone) [26]. In two RCTs involving 27,347 postmenopausal women, combined estrogen and progesterone were associated with a higher risk of breast cancer than the placebo. In contrast, estrogen alone among women with prior hysterectomy was associated with a lower risk of breast cancer than placebo [30]. In the present study, when analyzed according to types of HRT medication, combined estrogen and progesterone, estrogen alone, and tibolone were not risk factors for ten cancers in postmenopausal women with de novo or a history of endometriosis. Moreover, combined estrogen and progesterone reduced the risks of liver and thyroid cancers, and estrogen alone reduced the risks of breast and lung cancers (Table 3). In a prior study, estrogen alone was identified as a risk factor for ovarian cancer in postmenopausal women (*n* = 10,304) with de novo or a history of endometriosis administered HRT and registered by HIRA between 2008 and 2020 [31]. This difference was attributed to the very low incidence of ovarian cancer (present study: 0.4%; prior study: 0.3%), even though the present ovarian cancer dataset included more patients (*n* = 14,724) registered during a longer period (2008 to 2022) compared to a previous study [31].

In the present dataset for each cancer, the ages at the endometriosis diagnosis ranged from a mean of 48.4 to 48.5 years (Appendix A). Women who were diagnosed with endometriosis at an older age had an increased risk of colon, gastric, and lung cancers, which coincided with the fact that the incidence of those cancers increased with age based on the Korea Central Cancer Registry (KCCR), whereas women who were diagnosed with endometriosis at an older age had a decreased risk of breast cancer, which coincided with the fact that the incidence of this cancer decreased with age after the peak incidence at 45 to 49 years (KCCR) (Appendix A) [32]. Moreover, women who were diagnosed with endometriosis at an older age had an increased risk of cervical and ovarian cancers (Appendix A). However, the incidence of those cancers decreased with age after the peak incidence at 40 to 49 years and at 50 to 59 years, respectively (KCCR) [33,34].

In the present study, women who were more recently diagnosed with endometriosis had an increased risk of uterine and ovarian cancers, which coincided with a continued increase in the annual incidence of those cancers (KCCR), while women who were more recently diagnosed with endometriosis had a decreased risk of colon, gastric, liver, and lung cancers, which coincided with a continued decrease in the annual incidence of those cancers (KCCR) (Appendix A) [34,35,36]. However, the risk of cervical cancer, with a consistent decrease in annual incidence (KCCR), has increased for women diagnosed with endometriosis more recently (Appendix A) [33,35].

In a nationwide cohort study (*n* = 45,790), the incidences of endometrial and ovarian cancers increased with time after endometriosis diagnosis (standardized incidence ratios: endometrial cancer, 1.00 at one to four years and 1.37 at five to nine years post-diagnosis; ovarian cancer, 1.51 at one to four years and 1.78 at five to nine years post-diagnosis), but the incidence of breast cancer was not associated with the time since the endometriosis diagnosis [24]. In the present study, women who received HRT underwent more surgeries for endometriosis in the dataset for each cancer (Appendix A). In addition, the risks of uterine, ovarian, and thyroid cancers increased as the number of surgeries for endometriosis increased, suggesting that the risks of those cancers increased with time since endometriosis diagnosis (Appendix A).

Based on the guidelines for the management of endometriosis, combined endometriotic lesion resection and concomitant hysterectomy with or without BSO might reduce the recurrence rates of endometriosis compared to resection of endometriotic lesions alone [37]. In a nationwide cohort study (*n* = 155,972), the risk of colon cancer increased in women who underwent a hysterectomy alone compared to women who did not [38]. In the present dataset for each cancer, regardless of HRT use, a hysterectomy for benign disease reduced the risks of cervical, uterine, and ovarian cancers, suggesting that a hysterectomy has therapeutic effects via the removal of the uterus as well as reduction in the risk of endometriosis recurrence (Appendix A).

In the present dataset for each cancer, women who received HRT had more surgeries for endometriosis and were diagnosed with endometriosis or each cancer at a younger age, suggesting that they might have been more concerned with their health and have had more tests (Appendix A).

This large-scale, nationwide, population-based cohort study was the first to examine the impact of HRT on the occurrences of various cancers in postmenopausal women with de novo or a history of endometriosis. The limitations of this study are as follows. First, the diseases and treatments were defined using the diagnostic and prescription codes without reviewing the medical records. Hence, a few women with incorrect codes might have been misdefined. Second, the time of menopause and the severity of endometriosis could not be determined in the population because of the characteristics of claims data. Therefore, the date when women in the HRT group first received HRT was defined as the date at menopause in both groups. Third, women diagnosed and managed with endometriosis before the start of the HIRA dataset and subsequently underwent recurrence of endometriosis might have been included. Fourth, the significance of the impacts of HRT on occurrences of the ten cancers in postmenopausal women with de novo or a history of endometriosis is limited because of the very low incidence of each cancer and the very short duration of HRT. Finally, the impact of HRT on only ten cancers can be analyzed because of the extremely low incidences of many cancers.

## 5. Conclusions

This large-scale Korean nationwide cohort study showed that HRT does not increase the risks of nine cancers except uterine cancer alone in postmenopausal women with de novo or a history of endometriosis. Moreover, combined estrogen and progesterone have benefits on the risks of liver and thyroid cancers, and estrogen alone has benefits on the risks of breast and lung cancers. These results provide clinically useful information to guide the use of HRT in postmenopausal women with de novo or a history of endometriosis. Nevertheless, large-scale studies will be needed to confirm these results.

## Figures and Tables

**Figure 1 cancers-16-00809-f001:**
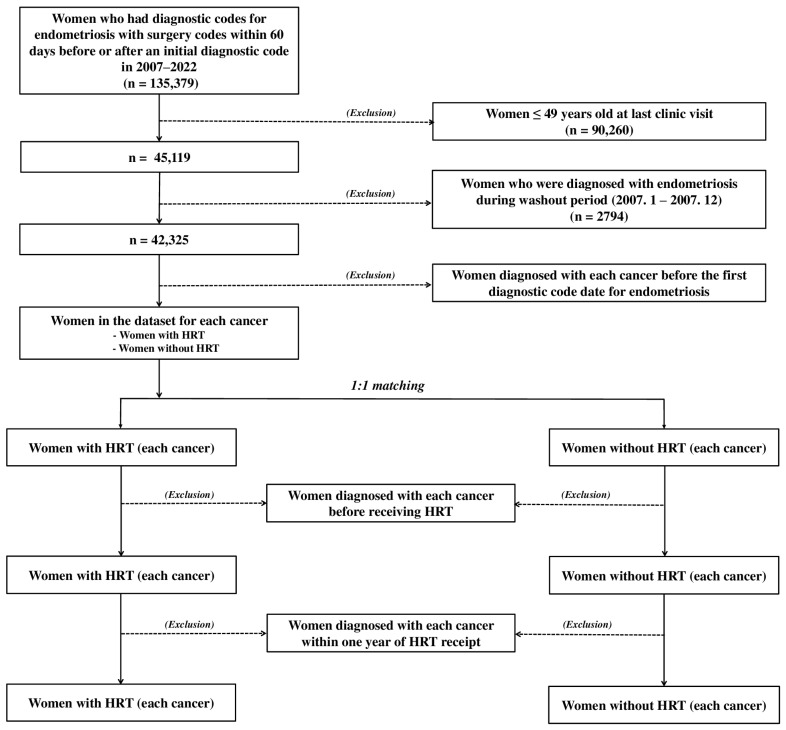
Flow chart for extracting eligible patients. The datasets for each cancer include a dataset for cervical cancer, a dataset for uterine cancer, a dataset for ovarian cancer, a dataset for breast cancer, a dataset for colon cancer, a dataset for gastric cancer, a dataset for liver cancer, a dataset for lung cancer, a dataset for pancreatic cancer, and a dataset for thyroid cancer.

**Table 1 cancers-16-00809-t001:** Characteristics of the datasets for various cancers related to HRT in women with postmenopausal endometriosis (HIRA claims data 2008–2022).

	Duration of HRT (Years)	Types of HRT Medication		
		Combined Estrogen and Progesterone	Estrogen Alone	Tibolone
Dataset for cervical cancer	1.3 ± 2.1	5093 (34.6)	7185 (48.9)	7551 (51.4)
Dataset for uterine cancer	1.3 ± 2.1	5077 (34.7)	7151 (48.8)	7526 (51.4)
Dataset for ovarian cancer	1.3 ± 2.1	5117 (34.7)	7197 (48.8)	7584 (51.4)
Dataset for breast cancer	1.3 ± 2.1	5069 (34.8)	7127 (48.9)	7490 (51.4)
Dataset for colon cancer	1.3 ± 2.1	5106 (34.6)	7190 (48.8)	7576 (51.4)
Dataset for gastric cancer	1.3 ± 2.1	5109 (34.7)	7181 (48.8)	7557 (51.3)
Dataset for liver cancer	1.3 ± 2.1	5098 (34.6)	7184 (48.8)	7576 (51.4)
Dataset for lung cancer	1.3 ± 2.1	5127 (34.6)	7233 (48.8)	7610 (51.4)
Dataset for pancreatic cancer	1.3 ± 2.1	5107 (34.7)	7180 (48.8)	7570 (51.4)
Dataset for thyroid cancer	1.3 ± 2.1	4927 (34.8)	6916 (48.8)	7289 (51.4)

HIRA, health insurance review and assessment service; HRT, hormone replacement therapy. All values are expressed as mean ± standard deviation or number (%).

**Table 2 cancers-16-00809-t002:** Incidences of various cancers according to HRT in women with postmenopausal endometriosis (HIRA claims data 2008–2022).

	Total (100.0%)	HRT (−) (50.0%)	HRT (+) (50.0%)	Adjusted OR (95% CI) ^a^	*p* Value ^a^
Cervical cancer					
Negative	29,334 (99.7)	14,688 (99.7)	14,666 (99.7)	ref	0.437
Positive	78 (0.3)	38 (0.3)	40 (0.3)	1.195 (0.762, 1.873)	
Uterine cancer					
Negative	29,126 (99.4)	14,567 (99.5)	14,559 (99.4)	ref	0.102
Positive	168 (0.6)	80 (0.6)	88 (0.6)	1.295 (0.95, 1.766)	
Ovarian cancer					
Negative	29,332 (99.6)	14,663 (99.6)	14,669 (99.6)	ref	0.94
Positive	116 (0.4)	61 (0.4)	55 (0.4)	0.986 (0.682, 1.426)	
Breast cancer					
Negative	28,754 (98.6)	14,389 (98.7)	14,365 (98.6)	ref	0.511
Positive	396 (1.4)	186 (1.3)	210 (1.4)	1.069 (0.876, 1.306)	
Colon cancer					
Negative	29,364 (99.6)	14,686 (99.7)	14,678 (99.6)	ref	0.676
Positive	106 (0.4)	49 (0.3)	57 (0.4)	1.086 (0.739, 1.595)	
Gastric cancer					
Negative	29,453 (99.8)	14,727 (99.8)	14,726 (99.8)	ref	0.884
Positive	65 (0.2)	32 (0.2)	33 (0.2)	0.964 (0.591, 1.572)	
Liver cancer					
Negative	29,328 (99.5)	14,662 (99.5)	14,666 (99.6)	ref	0.485
Positive	138 (0.5)	71 (0.5)	67 (0.5)	0.887 (0.634, 1.242)	
Lung cancer					
Negative	29,543 (99.7)	14,770 (99.7)	14,773 (99.7)	ref	0.47
Positive	81 (0.3)	42 (0.3)	39 (0.3)	0.851 (0.548, 1.32)	
Pancreatic cancer					
Negative	29,326 (99.6)	14,667 (99.6)	14,659 (99.6)	ref	0.666
Positive	122 (0.4)	57 (0.4)	65 (0.4)	1.082 (0.756, 1.548)	
Thyroid cancer					
Negative	27,986 (98.7)	14,017 (98.9)	13,969 (98.5)	ref	0.072
Positive	370 (1.3)	161 (1.1)	209 (1.5)	1.211 (0.983, 1.491)	

CI, confidence interval; HIRA, health insurance review and assessment service; HRT, hormone replacement therapy; OR, odds ratio; ref, reference. Values are expressed as numbers (%). ^a^ The data were adjusted for age at endometriosis diagnosis, year of endometriosis diagnosis, number of surgeries for endometriosis, and hysterectomy for benign disease.

**Table 3 cancers-16-00809-t003:** Associations of HRT with various cancer occurrences in women with postmenopausal endometriosis (HIRA claims data 2008–2022).

	Use of HRT		Duration of HRT per 1 Year		Types of HRT Medication					
					Combined Estrogen and Progesterone		Estrogen alone		Tibolone	
	Adjusted HR (95% CI) ^a^	*p* Value ^a^	Adjusted HR (95% CI) ^b^	*p* Value ^b^	Adjusted HR (95% CI) ^b^	*p* Value ^b^	Adjusted HR (95% CI) ^b^	*p* Value ^b^	Adjusted HR (95% CI) ^b^	*p* Value ^b^
Total (100.0%)										
Cervical cancer	1.377 (0.874, 2.169)	0.168								
Uterine cancer	1.542 (1.129, 2.106)	0.007								
Ovarian cancer	1.15 (0.791, 1.672)	0.465								
Breast cancer	1.058 (0.863, 1.297)	0.590								
Colon cancer	1.03 (0.697, 1.522)	0.883								
Gastric cancer	0.902 (0.55, 1.480)	0.684								
Liver cancer	0.831 (0.59, 1.172)	0.292								
Lung cancer	0.795 (0.508, 1.243)	0.314								
Pancreatic cancer	1.059 (0.735, 1.527)	0.759								
Thyroid cancer	1.187 (0.961, 1.467)	0.113								
HRT (+) (50.0%)										
Cervical cancer			0.818 (0.655, 1.022)	0.077	1.903 (0.922, 3.929)	0.082	1.053 (0.52, 2.134)	0.886	0.619 (0.284, 1.351)	0.229
Uterine cancer			0.944 (0.856, 1.042)	0.254	1.056 (0.651, 1.725)	0.816	0.728 (0.447, 1.186)	0.203	0.652 (0.386, 1.098)	0.108
Ovarian cancer			0.979 (0.876, 1.095)	0.714	0.987 (0.549, 1.774)	0.964	1.15 (0.627, 2.109)	0.652	0.933 (0.499, 1.742)	0.827
Breast cancer			0.977 (0.915, 1.044)	0.497	0.804 (0.581, 1.111)	0.186	0.715 (0.517, 0.987)	0.042	0.8 (0.566, 1.13)	0.205
Colon cancer			0.988 (0.894, 1.093)	0.819	1.713 (0.982, 2.989)	0.058	1.333 (0.761, 2.335)	0.315	1.47 (0.826, 2.615)	0.190
Gastric cancer			1.099 (0.986, 1.225)	0.088	0.885 (0.42, 1.864)	0.748	0.986 (0.458, 2.125)	0.972	0.990 (0.446, 2.201)	0.981
Liver cancer			0.378 (0.203, 0.702)	0.002	0.378 (0.203, 0.702)	0.002	1.052 (0.588, 1.883)	0.865	1.048 (0.571, 1.926)	0.879
Lung cancer			1.018 (0.9, 1.151)	0.782	1.204 (0.565, 2.579)	0.626	0.373 (0.174, 0.8)	0.011	0.823 (0.370, 1.831)	0.633
Pancreatic cancer			1.029 (0.933, 1.135)	0.57	0.693 (0.392, 1.226)	0.208	0.805 (0.453, 1.431)	0.460	0.772 (0.421, 1.415)	0.402
Thyroid cancer			0.984 (0.923, 1.049)	0.623	0.669 (0.484, 0.925)	0.015	0.971 (0.703, 1.341)	0.857	0.891 (0.634, 1.251)	0.504

CI, confidence interval; HIRA, health insurance review and assessment service; HR, hazard ratio; HRT, hormone replacement therapy. ^a^ The data were adjusted for age at endometriosis diagnosis, year of endometriosis diagnosis, number of surgeries for endometriosis, hysterectomy for benign disease, and HRT. ^b^ The data were adjusted for age at endometriosis diagnosis, year of endometriosis diagnosis, number of surgeries for endometriosis, hysterectomy for benign disease, duration of HRT, and types of HRT medication. The red fonts show the statistically significant data.

## Data Availability

The data that support the findings of this study are available from the Health Insurance Review and Assessment Service (HIRA), but restrictions apply to the availability of these data, which were used under license for the current study and so are not publicly available. Data are available from the authors upon reasonable request and with permission of the HIRA.

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
