# Peer review of "Hormone Replacement Therapy and Risks of Various Cancers in Postmenopausal Women with De Novo or a History of Endometriosis"

_cancers, 2024, doi:10.3390/cancers16040809_

Round 1
Reviewer 1 Report
Comments and Suggestions for Authors
The current manuscript describes potential, statistically significant enrichments of cancer incidences in edometriosis patients that received hormone replacement therapy (HRT+) versus those that did not (HRT-).
The main problem I have with this manscript relates to the presentation of the data, which is just simply difficult (and way too complex) to understand since the central elements shown in the manuscript are massive. One giant flow chart that contains a large number of data (Figure 1), and two extremely large, endless tables that spread over several pages but have no clearly recognizable highlights (results). Tbales 1 and 3 both contain a lot of ground data (size of the cohorts, age averages, length of HRT etc) that do not represent results - but should be considered background information.
SPECIFIC REQUESTS
Table 1 and 3 must be broken down in at least 2 categories: those data that describe the size of the cohorts, or patient groups, for different cancers - and then the actual results, related to any statistical enrichment of the incidences in the 2 groups that are compared for each tumor type (HRT+ versus HRT-).
Otherwise, table 1 and 3 are impossible to read and comprehend, and it makes little sense to have two gigantic tables going over several pages. Could some of this not simply be shown as supplemental data? Essentially, table 1 does not contain any interesting results and only confirms the successful selection of patient groups that can be meaningfully compared with each other, using statistics. The same applies to large parts of table 3, but then there are apparently some relevant data hidden in this massive table and those are very difficult to find.
The flow chart (Fig 1) has similar issues: Its just massive, and as such, the font sizes in the chart will probably be too small for the publication. I can understand that the authors try to implement all of the relevant data in a single file, but in printed form (or PDF) this will be very hard to read. And in printed form, even more. And as with Table 1 & 3 - how much of this is necessary? Much of it can be put into a large-scale supplementary file (could be Excel) that describes the composition of the patient groups and their specific traits, that have been compared by statistics.
Then, the results are largely negative, although this should not prohibit publication. Tables 2 and 3 both show that there is no very significant enrichment, or enhanced incidence, in patients that have received HRT versus those that did not. Apparently, according to the p-values, there isnt any enhanced risk.
And similar to table 1, in table 3 there are many data included that dont really adress the main question (= cancer risk, yes or no), but instead describe the ground data set for the many different patient cohorts in much detail. The question arises, once again, if this should be even shown in the main manuscript or if it is better off in supplemental data files? (Ideally, severy that can be clearly distiguished from each other). In my opinion, it doesnt make much sense to have gigantic tables that spread over several pages and are difficult to comprehend.
My suggestion is to condense the data, only show the most relevant stuff in the actual manuscript, and delegate other issues that describe the basic data structure to supplemental. And then, also do not forget to highlight the few statistically significant findings that may suggest consequences from HRT for cancer risk and incidence. This is missing in the current manuscript.
This makes it possible to read and understand the manuscript. Claims made by the authors like "Moreover, combined estrogen and progesterone reduced the risks of liver and thyroid cancers, and estrogen alone reduced the risks of breast and lung cancers (Table 3). [line 172-173]" will then become hopefully more transparent, and reproducible for the reader. Where in table 3 is this shown? Its completely overloaded with data to find the relevant things.
When these basic issues are resolved, the manuscript may become interesting enough for readers to be published. But in its current state, I do not see how its informative for readers.
Comments on the Quality of English Language
there are some issues with english language, but this is definitely not the main problem of the paper!
Author Response
I attached a file.

Reviewer 2 Report
Comments and Suggestions for Authors
The manuscript describes a Korean nationwide cohort study to investigate the impact of Hormone Replacement Therapy (HRT) on cancer risks in postmenopausal women with de-novo or a history of endometriosis. The study utilizes data from the National Health Service in South Korea from 2007 to 2022, focusing on ten different cancers. It evaluates the incidence and risk factors associated with each cancer, considering factors such as age, surgery for endometriosis, and HRT use. The results indicate that, overall, HRT does not increase the risks of nine cancers in this population, except for uterine cancer. Furthermore, the study suggests that combined estrogen and progesterone may have positive effects on reducing the risks of liver and thyroid cancers, while estrogen alone may reduce the risks of breast and lung cancers. The findings are considered clinically significant and provide insights into the use of HRT in postmenopausal women with endometriosis. While the manuscript provides a thorough examination of the impact of Hormone Replacement Therapy (HRT) on cancer risks in postmenopausal women with endometriosis, there are some aspects that are not explicitly addressed:
1) The study notes the very low incidence of certain cancers, which may limit the ability to draw robust conclusions for those specific cancer types.
2) The impact of HRT on cancer risks is evaluated based on a relatively short duration of HRT, Would the findings be limited by the short duration of exposure?
3) Any potential influence of different severities of endometriosis on the outcomes. Any information of the severity of endometriosis and its impact on cancer risks?
4) Besides the age, are there any considerations for ethnic or socioeconomic differences among the study participants? These factors could influence the generalizability of the findings.
5) Any alternative treatments for postmenopausal symptoms in women with endometriosis?
6) With Women in dataset for each cancer, if a woman has been diagnosed lung cancer, would she eligible for Dataset for the rest cancers except lung cancer? Or she will be excluded?
Minors:
Figure 1. For Women with HRT (each cancer) and Women without HRT (each cancer), both groups share the same N number, as indicated in the left and right textboxes.
Author Response
I attache a file.

Round 2
Reviewer 1 Report
Comments and Suggestions for Authors
The main concern with this manuscript related to the massive tables, which have now been outsourced largely to the supplemental table. Meanwhile, the tables that are still in the main manuscript are reasonable in size and should be more comprehensible for the readers (and us reviewers). What really drives the value of this shor manuscript is the large amount of retrospective data inn which a a considerable number of women are included. It is this vast size of the data set that also drives the statistics, and this could be stressed a bit more.
Its still not a massive amount of novel insights we are getting here, also since there is absolutely no discussion (or speculation) on the possible reasons for the reported observations. Even though some of this would be bordering on speculation, it may still be worthwhile adding this missing component to the discussion as every reader will ask himself or herself what may be behind these observations.
Comments on the Quality of English Language
this hasn't changed between the 2 versions and still needs some fixing, but in principle, I have seen much worse.
Author Response
I uploaded a file.

Reviewer 2 Report
Comments and Suggestions for Authors
The authors have addressed all my concerns and I have no further questions.
Round 3
Reviewer 1 Report
Comments and Suggestions for Authors
The authors have somewhat increased the volume of the discussion by a few issues and also added a small number of references, and although the paper is still a bit on the small/short side, I think it's a solid piece of information that hopefully will be interesting for readers active in this field of research.
Comments on the Quality of English Languagethere are still a few issues but nothing that cannot be resolved in production phase of the manuscript